# Measuring Awareness of Infection Control Guidelines for Patients with COVID-19 in Radiology Departments in Saudi Arabia

**DOI:** 10.3390/medsci9010018

**Published:** 2021-03-16

**Authors:** M. Almatari, Ali Alghamdi, Sultan Alamri, Mufeed Otaibey, Ahmad Joman Alghamdi, Azah Alasmari, Magbool Alelyani

**Affiliations:** 1Department of Physics, Faculty of Sciences, Al-Balqa Applied University, Al-Salt 19117, Jordan; matari@bau.edu.sa; 2Department of Radiological Sciences, Faculty of Applied Medical Sciences, Tabuk University, Tabuk 47713, Saudi Arabia; ah.alghamdi@ut.edu.sa; 3Department of Radiological Sciences, College of Applied Medical Sciences, Taif University, Taif 26521, Saudi Arabia; dr.ahmadjomanalghamdi@gmail.com; 4Department of Nuclear Medicine, King Abdullah Medical City, Makkah 24246, Saudi Arabia; mufeed_otaibey@hotmail.com; 5Department of Radiological Sciences, College of Applied Medical Sciences, King Khalid University, Abha 62529, Saudi Arabia; azasmari@kku.edu.sa (A.A.); maalalyani@kku.edu.sa (M.A.)

**Keywords:** COVID-19, medical imaging, radiology, infection control

## Abstract

Due to the contagious nature of the COVID-19 virus, healthcare workers are at a great risk of infection. Since medical imaging plays a significant part in the healthcare system and is often used in the diagnosis of potential COVID-19 patients, the radiology personnel are at a very high risk of becoming infected. Purpose: This study aims to assess the enforcement of infection control guidelines for patients with COVID-19 during medical imaging procedures and raise awareness of infection control in different hospitals in Saudi Arabia. Methods: A total of 128 responses were collected from four hospitals across Saudi Arabia using a new structured questionnaire, which was created for health workers by the WHO specifically for this purpose. Data were collected during the COVID-19 pandemic in April 2020. Results: Most participants correctly followed the guidelines of the WHO and Centers for Disease Control and Prevention (CDC) on infection control in the X-ray and general radiology departments. Guideline awareness was higher among magnetic resonance imaging (MRI) and computerised tomography (CT) radiographers, which reduced the risk of future infections. Out of the total respondents, 98.4% stated that they had received formal training in hand hygiene. Only 40.6% of participants, however, knew that respiratory droplets are the primary mode of transmission of the virus from person to person. Conclusion: The knowledge of healthcare professionals in the radiology department regarding infection control needs to be continually assessed. A focus on educational interventions on infection control is required in order to maintain well-informed medical staff.

## 1. Introduction

The focus of the world this year has been the outbreak of a novel coronavirus (COVID-19)—a respiratory disease that was first reported in Wuhan, China, and presents a substantial danger to international health. On March 11, 2020, the WHO declared COVID-19 a pandemic [1]. Similar to seasonal flu, COVID-19 is assumed to be spread among people in near proximity to one another via respiratory droplets expelled by an infected individual coughing or sneezing. Saudi Arabia, similarly to other countries, has been affected by this virus and the number of cases is increasing even though strict procedures are in place and the entire country is in lockdown. Due to the contagious nature of this disease, workers in healthcare sectors, such as hospitals, are at a high risk of infection with COVID-19 and the number of victims is rapidly increasing every day [2].

It has been necessary to educate frontline healthcare workers in order to slow and attempt to prevent the spread of this virus. The majority of cases amongst healthcare workers related to the early COVID-19 transmissions in China could be attributed to a lack of knowledge among medical staff about the prevention and control of contagious diseases [3]. The extremely contagious nature of the COVID-19 virus presents an extra risk for healthcare workers, in addition to the physical and psychological stresses they are enduring due to exhaustion and extended working hours [4].

Medical imaging plays a significant part in the healthcare system and has been used in the diagnosis of many potential COVID-19 patients. Computed tomography (CT) has been recognised by the World Health Organization (WHO) and in treatment guidelines from the National Health Commission (NHC) as a diagnosing and categorising tool for COVID-19 [5]. Suspected COVID-19 patients can undergo chest CT scans to confirm their case and classify it. These scans are, therefore, not only a screening tool, but also a measure of the degree of pulmonary association and of the course of the disease. Isolation and careful measures must be applied to protect both the staff and other patients inside the hospital or any medical facilities [6].

The radiology staff are at a great risk of infection from COVID-19. It is, therefore, essential to study the present infection control procedures in order to cut cross infection and protect healthcare workers in radiology departments, and to reinforce the need for additional training in infection control practices throughout major infectious disease epidemics. Guidelines for healthcare workers, particularly radiology staff, to avoid the exposure to COVID-19 include wearing personal protective equipment (PPE) all the time, keeping a distance of at least 1.8 m from patients, asking patients to wear a facemask, performing chest X-rays in the patients’ room with dedicated portable machines and only in emergency situations. Additionally, training staff is vital to reduce the spread of the disease. This includes infection control procedures and managing suspected or infected patients. As of May 2020, it is estimated that over 150,000 healthcare providers have been infected by COVID-19 and over 1413 deaths have been reported [7].

A recent study showed that only 40% of radiology department staff have the necessary knowledge of general infection control practice [8]. Five radiology staff in one hospital and several others in another were infected with COVID-19 due to the hospital’s failure to separate suspected patients into a designated area away from regular patients [9]. Radiology is one of the busiest departments in a hospital and so plays an essential role in developing policies and guidelines for said hospital, especially in the planning of treatment for patients based on their diagnosis, the management of patients, and the education and training for all personnel levels of responders in relevant procedures [10,11]. The radiology examination areas need to be managed to allow for specific COVID-19 timeslots to reduce any chance of the unnecessary spread of the virus [12].

The main aim of this study is to evaluate the implementation of infection control guidelines for patients with COVID-19 during medical imaging procedures and raise awareness of infection control in different hospitals in Saudi Arabia. This has been achieved through an online questionnaire that was created based on current guidelines and information for healthcare workers provided by the WHO and CDC.

## 2. Materials and Methods

### 2.1. Patients

The questionnaire was created using an online tool and was distributed through social media targeting all healthcare workers in the radiology departments in Saudi Arabia. The survey commenced in April 2020 and a total of 128 responses were received from healthcare workers in radiology departments. The professions of the people surveyed included radiologists, radiographers, radiographer assistants, medical physicists and nurses.

The self-administered questionnaire began with the hospital name, city, occupation, location within the radiology department (e.g., X-ray, ultrasound, nuclear medicine, CT scan and MRI) and socio-demographic questions such as age and gender, followed by 15 questions covering essential knowledge on infection control practices in their respective radiology departments. Many of these questions were taken from an online survey originally created to measure COVID-19 awareness between healthcare students and professionals in Mumbai [13], as well as from the present acting guidance and information for healthcare workers published by the CDC and WHO regarding infection control [12,14].

For data collection, a convenient sampling method along with frequency and percentages were used for the classification of responses. Subgroup division was based on gender, age groups (18–30), (31–45), and (>45) years and occupation inside the radiology department (radiologist, radiographer, radiographer assistant, medical physicist, and nurse). Training acknowledged by the participants for infection control procedures in the last three years was recorded. Information about the process was shared with the participants at the beginning of the study. Ethical approval was obtained from the Institutional Review Board (IRB) committees in the directorate of health affairs, Taif University (under registration number with KACST, KSA: HAP-02-T-067). Completing the questionnaire was considered as an implicit agreement to take part in the study; no names were requested from participants. To protect the privacy of participants, they were guaranteed that all related material would be saved and accessed by researchers only.

### 2.2. Data Analysis

Data collected from the questionnaires was exported to Microsoft Excel and then transferred to the Statistical Package for the Social Sciences (SPSS), version 23 (SPSS Inc., Chicago, IL, USA). Statistics, such as percentages and frequencies, were used to describe the participants’ demographic characteristics and responses. Categorical variables were evaluated using the chi-squared test as the significance level was set to be a *p*-value lower than 0.05. The rates of correct answers were compared and the association between the total correct answers and demographic characteristics was calculated using the ANOVA test.

## 3. Results

A total of 128 participants completed the questionnaire. Most importantly, among the final sample, a few of the demographic data were as follows: 46 (36%) were women, 75 (58.6%) were aged between 30 to 45, 111 (87.7%) were radiographers, 54 (42.8%) worked in the X-ray department and 22 (17.2%) were from Makkah. Other demographic data are shown in Table 1. Figure 1 shows the distribution of respondents according to their place of work. The most populous group worked in X-ray departments (54 participants (42.2%)), and the smallest group worked in ultrasound departments (four participants (3.1%)).

Of the total respondents, 98.4% stated that they had received formal training in hand hygiene in the past three years. Only 40.6% of participants knew that respiratory droplets are the primary mode of transmission of the virus from person to person. When participants were asked about the indicators for a patient who reported illness with symptoms ranging from mild to severe, 120 (93.8%) of the participants knew the main indicators, which include being in close contact or residing with an infected person or having travelled to a country where there is an ongoing spread of COVID-19. Regarding hand hygiene actions that prevent transmission of the virus to healthcare workers, 117 (91.4%) stated correctly that hand washing should be done after touching a patient and immediately after exposure to bodily fluids. This should also be done after exposure to the immediate surroundings of the patient or before putting on and upon removal of personal protective equipment (PPE), including gloves. A total of 100 (78.1%) of the respondents correctly stated that rubbing hands with soap and water for at least 20 s is the preferred method of hand hygiene for visibly soiled hands.

Additionally, 114 (89.1%) of the respondents correctly stated that the use of a face mask with people who are well was not essential to protect themselves from COVID-19 infection. When participants were asked about the most effective method for prevention of COVID-19 infection in the healthcare setting, 124 (96.9%) knew that avoiding exposure when caring for patients with confirmed or possible COVID-19 is better than having a vaccination, when available. Of the total sample, 126 (98.4%) stated that all personal protective equipment (gloves, gown, protective eyeglasses, respirator—N95 mask) must be worn when transporting a patient who was confirmed to have COVID-19. Additionally, 88 (86.8%) of the participants stated that all PPE must be worn when providing services to patients with a history of contact to COVID-19 who are being assessed for a non-infectious illness (e.g., hypertension or hyperglycemia). Regarding recommended infection control measures upon the arrival of a patient with suspected COVID-19 infection to the hospital, 117 (91.4%) of the participants stated that the following procedure must be followed: rapid triage for patients, application of respiratory hygiene and cough etiquette, and assignment in a separate, well-ventilated space that allows waiting symptomatic patients to be separated by six or more feet. Table 2 shows the distribution of participants’ correct answers for all questions.

The ANOVA test was used to identify the association of the mean scores of correct answers across genders, age groups, occupation and place of work (Table 3) in which mean, STD (standard deviation), and *p*-values are represented. Generally, there was no significant association between groups *p* > 0.05. Female participants for example, achieved higher scores (mean = 11.0, STD = ± 1.5) than males did (mean = 10.9, STD = ± 1.7) *p* = 0.546. Regarding age groups, older employees (>45 years) achieved the highest scores (mean = 11.7, STD = ± 1.7) *p* = 0.128. Radiologists had more knowledge than radiographers and medical physicists, *p* = 0.79, and nuclear medicine specialists achieved the highest scores compared with other modalities. The multiple comparisons between elements within groups, however, show significant associations within some categories. For example, participants aged >45 years were significantly more knowledgeable (mean = 11.7, STD = 1.1) than those aged 30–45 years (mean = 10.8, STD = 1.6) with *p* = 0.044. Additionally, radiologists achieved significantly higher scores (mean = 11.9, STD = 1.4) than radiographers (mean = 10.8, STD = 1.6) with *p* = 0.037. The rate of correct answers among participants ranged between 29.6 and 100%. The mean score of complete knowledge regarding COVID-19 was 10.9 (SD = 1.6, range: 4–14). Of all participants, only 0.8% achieved less than five correct answers, 40.6% answered between five and ten correct answers, and 58.6% had more than ten correct answers out of a total of 15 questions, which reflects poor, medium and excellent knowledge, respectively, in Figure 2.

## 4. Discussion

COVID-19 is spreading quickly around the world, affecting both global health services and economies. Radiology employees are in close contact with COVID-19 patients and thus, their knowledge of infection control is crucial to attempt to prevent or slow down the spread of the virus. Overall, the findings presented in this study show that the radiology staff have good knowledge of infection control guidelines. They showed a lack of knowledge, however, in the practical side of infection control, as indicated by lower proportions of correct answers for questions 4, 8, 10, 12, 13, and 14.

Although the majority of staff working in the radiology department showed a good understanding of the symptoms of the virus and the importance of hygiene in preventing its spread (94 and 91%, respectively), only 41% of them knew that the primary mode of transmission of the virus is from one person to another. These findings are consistent with related studies with other healthcare providers [13,15]. The relationship between knowledge about infection control and the age of the participants was not significant and the results showed a similar trend to that of a study by Asaad et al. (2018), where older participants showed higher rates of sufficient knowledge [16]. Similar to what has been reported in the literature, this study agreed that radiologists had more knowledge about infection control than radiographers did [15,16,17,18]. More educational programs should focus on non-physicians, therefore, to close the gap in knowledge between radiologists and other radiology staff.

Correct practices of hand hygiene like hand washing and hand rubbing with alcohol-based gels for at least 20 seconds are essential in preventing the spread of this virus. Although the radiology staff in this study had more knowledge about hand hygiene 76% than 51% in the study of Modi et al. [13], this percentage is still low and may lead to severe consequences. Approximately 89% of the sample participants of this study believed that the use of a face mask should be limited to only those suspected of being ill with the virus and healthcare professionals. Even though there is evidence that wearing masks is necessary for all people, whether or not they are infected, the results of this study are in accordance with the recommendations of all respectable health organisations, as there is a need to save the limited supplies of facemasks for patients and healthcare providers [19].

In this study, participants showed excellent knowledge of the type of PPE that should be worn when coming into contact with a patient with COVID-19 (98%); however, they wrongly considered it to be safe to be in contact with asymptomatic patients. This is a common gap in knowledge, which has been reported in several studies [11,20,21]. Unfortunately, only 73% of the participants in this study knew that the collection of respiratory specimens should be done in an airborne infection isolation room that is equipped with high-efficiency filters and is kept under negative pressure.

Almost 81% of the respondents thought that any non-urgent imaging procedures should be delayed until the disease is considered non-contagious. This is consistent with the policies and guidelines for COVID-19 preparedness in order to limit the general population exposure to the hospital environment as well as to preserve resources and hospital beds [10]. Additionally, 33% of the participants falsely thought that imaging inpatients should be performed in the main rooms. It is essential to perform imaging in rooms set aside for suspected COVID-19 patients to minimize the contact of the inpatients with the general population. Moreover, only 52% thought that it was recommended to perform imaging in the patient’s room while having the imaging source outside. This has been recommended in the literature to reduce exposure, use of PPE, and cleaning times [11].

## 5. Conclusions

In conclusion, upon the date of submission of this paper, this is the first study to assess the awareness of COVID-19 infection control guidelines among radiology staff in Saudi Arabia. Due to the close contact they have with patients and the vital role radiology plays in the health system, increased knowledge of radiology staff about infection control can prevent the spread of the disease. The radiology department healthcare workers in Saudi Arabia showed sufficient awareness of COVID-19 in the healthcare setting with an overall acceptable level of correct answers according to the international and local guidelines. A higher percentage of correct responses was received from radiologists and nurses, and the lowest score was from radiographers and medical physicists. This study shows the need for continuous education and training programs on infection control targeting all healthcare workers.

## Figures and Tables

**Figure 1 medsci-09-00018-f001:**
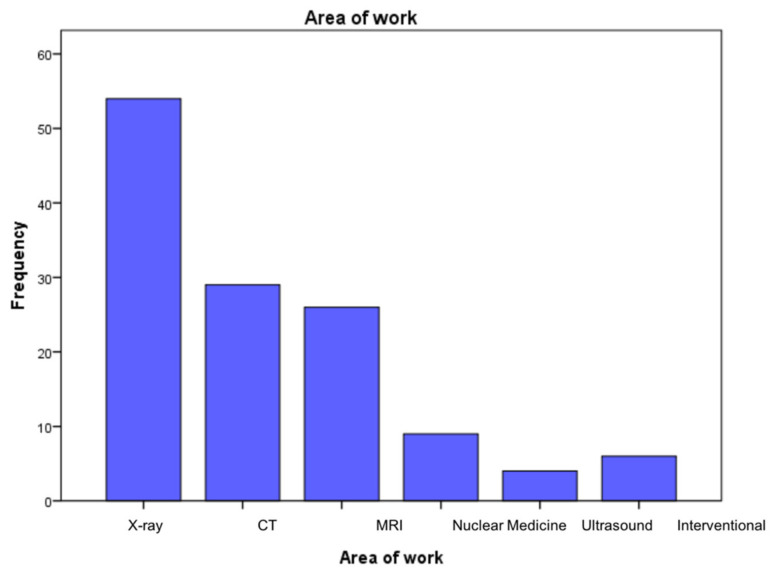
The distribution of participants according to the place of work.

**Figure 2 medsci-09-00018-f002:**
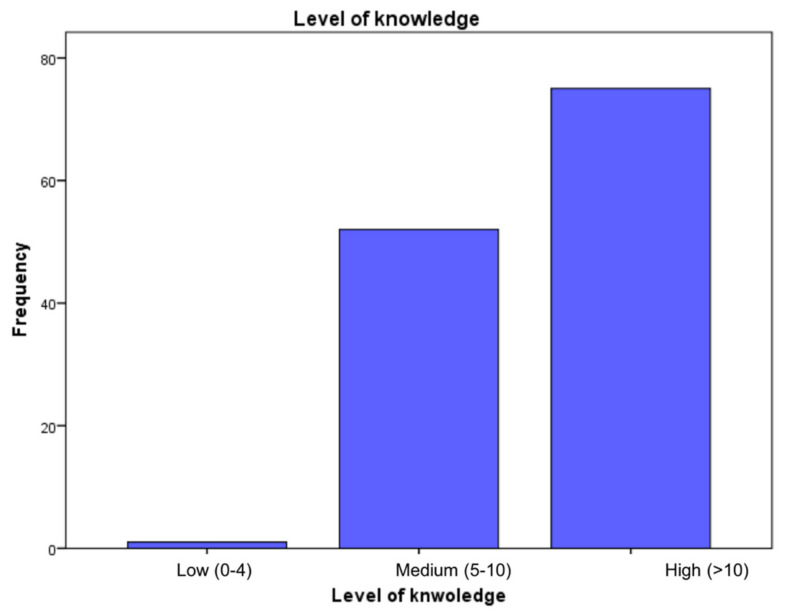
Level of knowledge among overall participants measured by the number of correct answers out of 14 questions (*n* = 128).

**Table 1 medsci-09-00018-t001:** Demographic characteristics of participants (*n* = 128).

Variable	Category	Number	Percent %
Gender	Male	82	64
	Female	46	36
Age	18–29	38	29.7
	30–45	75	58.6
	>45	15	11.7
Occupation	Radiologist	10	7.8
	Radiographer	111	87.7
	Physicist	7	5.5
Place of work	X-ray	54	42.8
	CT	29	22.7
	MRI	26	20.3
	Nuclear Medicine	9	7
	Ultrasound	4	3.1
	Interventional Radiology	6	4.7
City	Riyadh	3	2.3
	Makkah	22	17.2
	Dammam	16	12.5
	Taif	19	14.8
	Tabuk	42	32.8
	Albaha	4	3.1
	Abha	7	5.5
	Jazan	9	7.1
	Hail	6	4.7

**Table 2 medsci-09-00018-t002:** The distribution of correct answers for all questions.

Questions	Correct Answers
1. The main mode of transmission of virus from person to person	40.6%
2. Reported illnesses have ranged from mild to severe symptoms of cough, fever, breathlessness which can appear 2–14 days after exposure. For which of the following situations is medical advice indicated?	93.8%
3. Which of the following hand hygiene actions prevents transmission of the virus to the healthcare worker?	91.4%
4. Preferred method of hand hygiene for visibly soiled hands	78.1%
5. Use of a face mask is not essential in which of the following groups?	89.1%
6. Which of the following is the most effective method for prevention of COVID-19 infection in the healthcare setting?	96.9%
7. What personal protective equipment (PPE) should be worn by individuals transporting patients who are confirmed with or under investigation for COVID-19 within a healthcare facility?	98.4%
8. What PPE should be worn by healthcare practitioners (HCPs) providing care to asymptomatic patients with a history of exposure to COVID-19 who are being evaluated for a non-infectious complaint (e.g., hypertension or hyperglycemia)?	68.8%
9. Which of the following are recommended infection control measures upon arrival of a patient with suspected COVID-19 infection?	91.4%
10. A recommended infection prevention and control measure is to perform aerosol-generating procedures, including collection of diagnostic respiratory specimens, in an AIIR (Airborne Infection Isolation Room).	73.4%
11. Non-emergent procedures should be postponed or cancelled during the pandemic surge; including mammography, lung cancer screening, and CT colonoscopy as well as DEXA	80.5%
12. If portable imaging is possible, inpatient is imaged (e.g., chest X-ray) using main rooms.	77.3%
13. Imaging in patients with confirmed or suspected COVID-19, radiographic imaging can be performed through the glass of isolation rooms, where only the cassette enters the room and the source remains outside	51.6%
14. In case of droplet/contact, post imaging room cleaning should be performed using standard antiseptic wipes and there is no need for room closure if there is adequate air circulation in the room	61.7%

**Table 3 medsci-09-00018-t003:** The rates of correct answers according to demographic characteristics (*n* = 128). The mean difference is significant at the 0.05 level.

Variable	Category	Mean Score (out of 15)	Std ±	*p*-Value
Gender	Male	10.9	1.7	0.546
	Female	11	1.5	
Age	18–29	10.9	1.5	0.128
	30–45	10.8	1.7	
	>45	11.7	1.1	
Occupation	Radiologist	11.9	1.4	0.079
	Radiographer	10.8	1.6	
	Physicist	11.4	1.9	
Place of Work	X-ray	11.1	1.8	0.513
	CT	10.7	1.2	
	MRI	10.5	1.4	
	Nuclear Medicine	11.6	1.9	
	Ultrasound	11	1.8	
	Interventional Radiology	11.3	1.2	
City	Riyadh	9.6	0.5	0.412
	Makkah	10.8	1.6	
	Dammam	11.7	1.4	
	Taif	10.9	1.3	
	Tabuk	10.7	1.9	
	Albaha	11.7	1.3	
	Abha	10.8	1.2	
	Jazan	10.7	0.8	
	Hail	10.5	1	

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
