# Peer review of "Measuring Awareness of Infection Control Guidelines for Patients with COVID-19 in Radiology Departments in Saudi Arabia"

_medsci, 2021, doi:10.3390/medsci9010018_

Round 1

Reviewer 1 Report

This is an interesting and useful paper on infection control awareness in radiology departments during the COVID 19 pandemic.

However, as the study was conducted almost a year ago (April 2020) and given the rate new information comes into light, I feel the way it is presented is a bit outdated.

In order to still be able to use the information gathered, I would try to make a connection with the COVID 19 situation today in the country, e.g. referring to expected number of cases in June 2020 (line 188) is not of much use.

I also propose to enrich the paper with some additional information a) on the existence of guidelines for healthcare professionals, the way these are endorsed and applied in hospitals in Saudi Arabia, b) on the conduct of training on COVID 19 in hospital workers since the study as suggested, c) on the SARS-CoV-2 infections recorded among hospital staff since the study and d) on the conduct of and participation in possible preparedness activities that have taken place in the hospitals.

Some English editing is also necessary e.g line 62 hospital staff not staffs, line 153 suggestive patients, what does this mean, line 190 slow down the spread.

Reviewer 2 Report

Reviewer’s Comments

Name of journal: Medical Sciences (ISSN 2076-3271)

Manuscript ID: medsci-1133066

Type: Article

Title: Measuring Awareness of Infection Control Guidelines for Patients with COVID-19 in Radiology Departments in Saudi Arabia

Authors: M. Almatari , Ali Alghamdi , Sultan Alamri * , Mufeed Otaibey , Magbool Alelyani

Reviewer

Summary:

The following study evaluates the application of the guidelines, for the control of COVID-19 infections, through a structured questionnaire, in the general radiology departments of four hospitals in Saudi Arabia. This study evaluates awareness of guidelines and infection control among healthcare professionals. Concluding that it is necessary to evaluate the knowledge of health professionals and improve educational interventions on infection control, in order to avoid an increase in COVID-19 infection. The study presented deals with an excellent topic of prevention and hygiene applied in health facilities and the awareness that health personnel need in order to face the COVID-19 epidemic. The study cohort could be increased, taking into account other wards or other hospitals and finally assessing the percentage of infected health workers according to the awareness of the guidelines and control of the infection. This would allow for an informed program of better guidelines to be followed and taken as an example to minimize the incidence of infection in hospital units.

Major issues:

1) The study cohort could be increased, taking into account other wards or other hospitals and finally assessing the percentage of infected health workers according to the awareness of the guidelines and control of the infection. This would allow for an informed program of better guidelines to be followed and taken as an example to minimize the incidence of infection in hospital units.

Minor issues:

1) Introduction, line 44  : « such as hospitals, are at a high risk », please check the space after the comma.

2) Line 54 : « CT has been recognised… », please insert the extended form in parentheses.

3) Materials and Methods, line 103 : « was recorded. Information », please check the space after the comma.

4) Line 155: « be a p-value lower than .05 » please check and correct this value. The same at line 160, and check the next lines, in according to table 3.

5) Discussion, line 200 : « the participants was not significant and », please correct it.

6) Line 202 : « ge [15]. Similar to », please check the space after the comma, and check throughout the text.

7) Line 210: « hand hygiene (76%) 209 than in the study of Modi et al. (2020) (51%) [12] » it is better to correct it to « hand hygiene 76% than 51% in the study of Modi et al. [12]»
